# Transpilers: A Systematic Mapping Review of Their Usage in Research and Industry

**Andrés Bastidas Fuertes** [1,*][ID]**, María Pérez** [1][ID] **and Jaime Meza Hormaza** [2][ID]

[1] Facultad de Ingeniería en Sistemas, Escuela Politécnica Nacional, Quito 170525, Ecuador; maria.perez@epn.edu.ec

[2] Facultad de Ciencias Informáticas, Universidad Técnica de Manabí, Portoviejo 130105, Ecuador; jaime.meza@utm.edu.ec

[*] Correspondence: andres.bastidas02@epn.edu.ec or andres.bastidas@smartwork.com.ec

**Abstract:** Transpilers refer to a special type of compilation that takes source code and translates it into target source code. This type of technique has been used for different types of implementations in scientific studies. A review of the research areas related to the use of transpilers allows the understanding of the direction in this branch of knowledge. The objective was to carry out an exhaustive and extended mapping of the usage and implementation of transpilers in research studies in the last 10 years. A systematic mapping review was carried out for answering the 5 research questions proposed. The PSALSAR method is used as a guide to the steps needed for the review. In total, from 1181 articles collected, 683 primary studies were selected, reviewed, and analyzed. Proposals from the industry were also analyzed. A new method for automatic data tabulation has been proposed for the mapping objective, using a relational database and SQL language. It was identified that the most common uses of transpilers are related to performance optimizations, parallel programming, embedded systems, compilers, testing, AI, graphics, and software development. In conclusion, it was possible to determine the extent and identification of research sub-areas and their impact on the usage of the transpilers. Future research could be considered about the usage of transpilers in transactional software, migration strategies for legacy systems, AI, math, multiplatform games and apps, automatic source code generation, and networking.

**Keywords:** transpiler; source-to-source compiler; transcompiler; cross compiler; software architecture; systematic literature review





## 1. Introduction

### 1.1. Background

Hirzel et al. [1] define the transpilation process as the one in which software is written in a source language, but compiled to and executed in a different programming language. Chaber et al. [2] explain that transpilation is a code generation method in which a translation occurs from one high-level language to another low-level language.

The first uses of transpilers date back to the 1970s and 1980s. In 1978, Intel proposed an automatic code translator to convert 8-bit programs to their equivalent 16-bit programs [3]. The XLT86™ was proposed, a 8080 to 8086 Assembly Language Translator in 1981. Its goal was to automatically transform ASM type files to A86 type files [4].

Transpilers are quite broad in their applications and use, for example, DMS, an approach to managing software scalability and evolution [5], direct translation from Python to Julia [6], from C to Rust [7], from Java to Javascript [8], or code parallelization [9], embedded optimizations [10], and many others.

Alike other technologies, transpilers have been improved along with the evolution of programming languages and the appearance of new technologies and devices. An example of its recent use is the creation of the Typescript language (https://www.typescriptlang.org

(accessed on 5 January 2023)), which becomes an extension of the Javascript language, adding a syntax based on types, classes, modules, and other features [11]. Typescript is widely used alone and as the programming language for frameworks such as Angular (https://angular.io/ (accessed on 5 January 2023)), ReactJS/React Native (https://reactjs.org/ (accessed on 5 January 2023)), NativeScript (https://nativescript.org/ (accessed on 5 January 2023)), Redux (https://redux.js.org/ (accessed on 5 January 2023)), and others.

Due to the transpilers' source code translation and rewriting capabilities, transpilers have been used for several different applications, such as automatic code refactoring, source code optimizations, porting, platform compatibility preparation, front-end or gaming frameworks, and so on.

An entire-topic mapping review about transpilers' usage in research last years, and also along with main applications in the industry; is considered significant because it allows for building a comprehensive overview of the usage of this technology. It also allows us to expose the categorization of recent works, recognize industry applications, and identify possible new related research areas. Those are the intended contributions of this work.

Consequently, a review of all the studies that have the application of transpilers in the field of computer science in the last years in scientific databases was considered. It was designed as a wide-field review of the main topic, without applying specific segmentation or filtering toward a particular sub-specialty or any kind of application. This allows a broad spectrum mapping of its usage in research.

Regarding industry applications of transpilers, we carried out an exploratory search using public search engines, frameworks documentation, and cross-reference search. This was performed as a complementary study, with the intent to show the existence of several frameworks and platforms that use transpilers inside generally available products.

Since there were a large number of articles that needed to be analyzed, this work also involved the proposal of a new method for data extraction and classification. Formats were extracted from scientific databases to make automated term tabulation using an SQL database. This method allows the processing of the articles' metadata for semi-automatic classification and semi-automated data synthesis.

### 1.2. Transpiler Disambiguation

In the scope of Computer Science and software architecture, a generic definition is proposed in this paper. "A transpiler is a tool designed to automatically transform source code made up with a source high-level programming language, into another source code made with a target high-level programming language, which should be algorithmic-equivalent".

Despite "transpiler" being the term used in this work, there are several terms that are related to the former and all of them refer to the same concept. In this work, they will be treated as synonyms. They are the following: transpiler, transcompiler, source-to-source compiler, s2s compiler, and cross-compiler. Despite several parts of this document using the term "translation", it is not considered formally as a synonym because it could be confused with other non-computer science research areas such as linguistics or education.

In reference to automatic code generation techniques or template-based software generation, and even software generation based on artificial intelligence techniques, although they are techniques that have similar procedures or results, their technique and application are different from transpilers, since there is no source code transformation process. Those are not considered inside the scope of the term definition in this article.

### 1.3. Objectives

The main objective of this study is to have a systematic mapping review of the use of transpilers in the computer science area in the last 10 years, so as to examine the extent, scope, and nature of the research activity, along with its usage.

The following are presented as secondary objectives:

- Identify the areas, approaches, or specific topics where transpilers have been used the most;
- Determine the use of transpilers in the construction of business applications and specifically regarding their use in front-end and back-end frameworks;
- Determine the use of transpilers in the framework of the approach to patterns and designs of software architecture.

### 1.4. Research Questions

The following are the research questions for the present study:

- RQ1: What are the descriptive statistics of the publications among selected articles?
- RQ2: In which scenarios are transpilers most used?
- RQ3: In which kind of industry applications are transpilers commonly used?
- RQ4: Which programming languages and technologies are mainly related to transpilers?
- RQ5: Which are the usages of transpilers in the scope of the back-end of a transactional application?

### 1.5. Content Organization

This article is organized according to Figure 1. The first section (the current one) presents the background, the disambiguation of the term "Transpiler", and related works. Section II presents the materials and methods that were used in the present study, using the steps of the PSALSAR method. Section III has the presentation of the results and the threats to validity. Section IV presents the interpretation and discussion of the results. Section V presents the conclusions and possible future work.

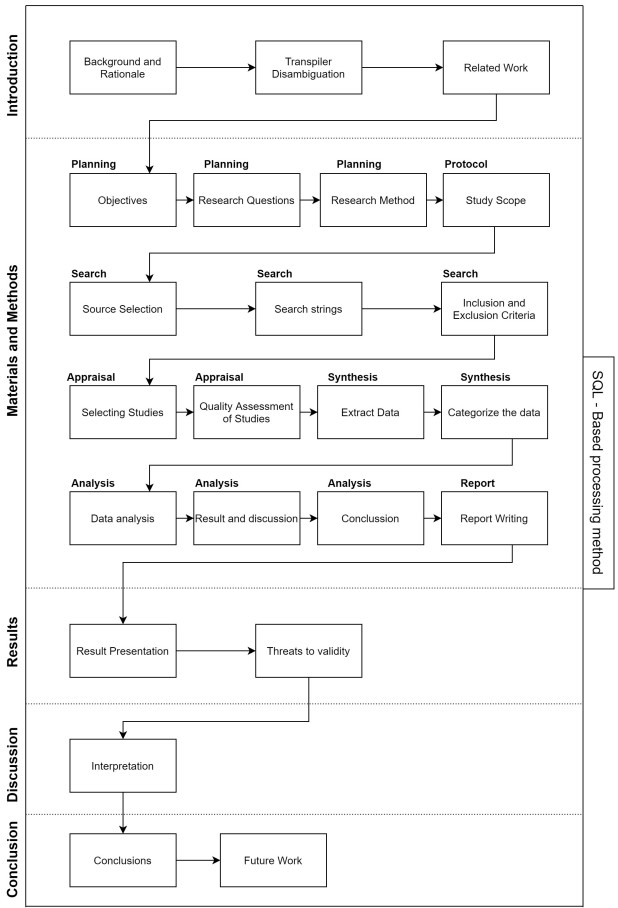

**Figure 1.** Article content structure.

## 2. Methods and Execution

### 2.1. Related Work

As a preamble to the execution of this study, a rapid literature review was carried out [12] with the specific objective of validating whether there is another article that has addressed the same approach or that has approaches related to the present study. This preliminary review allows us to identify which articles have been carried out previously and, based on their deepening, direct and enrich the result of the present investigation.

For this rapid literature review, the scientific databases Scopus, IEEE Xplore, ACM, and Springer Link were used. The following search string was considered: "("systematic literature review" OR "mapping study" OR "meta-analysis") AND ("transpiler" OR "transcompiler" OR "source to source compiler" OR "s2s compiler")" . Filters related to the year of the article were not placed. Additional filters were not considered in the search. The established selection criteria seeks to select the studies that, reading their title and abstract, could have a similar focus to the present study. Articles that do not present a literature review or mapping study will not be considered.

The search was executed and resulted in the following findings:

- Scopus: 7 articles;
- IEEE Xplore: 1 article;
- ACM: 6 articles;
- SpringerLink: 17 articles.

Starting from the articles obtained, we selected the articles that are closest to the main objective of this study. From this, six articles can be highlighted, and their interpretation is presented below:

- A Survey of Performance Tuning Techniques and Tools for Parallel Applications: [13]
  In this article, it is possible to see a survey related to tuning techniques for parallel applications. It is possible to see that one of the uses of transpilers is indeed in optimization for parallelization of source code. However, its objective has a specific approach to this topic, when the present article is focused on a mapping of the general scope of the use of transpilers.
- Evolution of Artificial Intelligence Programming Languages—a Systematic Literature Review: [14]
  In this article, the author proposes a literature review focused on the programming languages used for artificial intelligence (AI). Its objective has a different approach to the objective of the present study.
- Systematic mapping study of architectural trends in Latin America: [15]
  This article proposes a literature review focused on the uses of software architecture that have been proposed in Latin America. Although the use of transpilers is an element that is part of several software architecture proposals (Example: [16]), the present study has the objective of identifying the uses of transpilers, specifically, in any of their uses and without a defined geographical limitation.
- A systematic review on Transpiler usage for Transaction-Oriented Applications: [17]
  This article proposes a review of the specific literature for the use of transpilers in transaction-oriented applications. This article seeks to cover the use of transpilers in all its fields of use in recent years.
- Negative Perceptions About the Applicability of Source-to-Source Compilers in HPC: A Literature Review: [18]
  This article presents a study on the negative impacts of using a transpiler for HPC (high-performance computing). The focus of this article is specific to the stated objective, while the current article focuses on a wide coverage review of the use of transpilers in different topics.
- Using meta-heuristics and machine learning for software optimization of parallel computing systems: a systematic literature review: [19]

This article presents a literature review focused on methods to optimize software in parallel computing systems. Although one of the techniques to optimize parallel computing is the use of transpilers, it does not focus on other uses of this technology.

To the knowledge of the authors, there is no other article that has the same approach and objectives as the current work.

There are articles [20–23] that carry out systematic mapping reviews in the field of computer science, which propose comparative references of the type of study and expected results in the research.

## 2.2. Planning

### Research Method

The PSALSAR method is proposed by Mengist et al. [24] as a process design that allows the execution of systematic literature reviews and meta-analysis studies while allowing the generation of new knowledge, determining trends, and identifying existing gaps in specific topics. It corresponds to the fusion of the PRISMA method and the SALSA method. This method was proposed mainly for research in environmental sciences, but its authors mention that it can be used by anyone who wishes to conduct this type of study. It has been found that the method allows for a systematic and comprehensive review, making it suitable for the objectives of the present study.

The Table 1 presents an abstract of the PSALSAR method steps.

**Table 1.** PSALSAR steps.

| Step | Description |
| --- | --- |
| Procotol | For defining the study scope |
| Search | For defining the search strategy |
| Appraisal | For selecting the studies and to make quality assessment |
| Synthesis | For extracting and categorizing data |
| Analysis | For data analysis, results formulation, and discussion |
| Report | For defining conclusions and writing the report |

Next in this document, the steps of PSALSAR method will be presented as sections, showing the specific treatment of each one for this study.

## 2.3. Protocol

### Study Scope

The scope of the present study was defined as a general mapping of the use of transpilers in scientific databases. This condition meant that specific filters were not proposed for focusing on defined topics, but rather a general search was carried out on the use of transpilers in any scenario, provided that they are in the field of computer science for the last 10 years.

In addition, for specifying the scope of the current study, in line with the objectives and research questions, the use of the PICOC criteria (population, intervention, comparison, outcome, and context) [25] has been found appropriate for defining the global scope of search string terms, and the other search criteria. Table 2 presents the details of PICOC definitions for this work.

**Table 2.** PICOC details.

| Element | Definition | Synonyms |
|---|---|---|
| Population | Software architects, software developers, and software and computer science researchers | Computer science area |
| Intervention | Transpilers usage | • Transpiler—Transpilation<br>• Transcompiler—Transcompilation<br>• Cross Compiler—Cross Compilation<br>• Source to Source Compiler—Source to Source Compilation<br>• S2S Compiler—S2S Compilation |
| Comparison | n/a | n/a |
| Outcome | Scenarios of transpiler usages in scientific articles | n/a |
| Context | All recent studies | The period of the last 10 years |

*2.4. Search*

2.4.1. Source Selection

There are several scientific sources that could be selected for the current study in order to find their coverage and topicality. For the source selection, the following criteria were considered:

- Recognized or highly cited scientific databases or search systems
- Databases that are commonly used for computer science publications
- Consider a curated catalog of information, not a crawler-based search engine
- The capacity to include advanced search criteria to fit the scope of the current study
- The use of principal databases and also supplementary specialized databases

Gusenbauer et al. propose a method to evaluate which academic search systems are suitable for systematic reviews [26]. They present a chart for evaluating 28 academic search systems in order to classify them using different evaluation requirements. From these, four academic search systems were selected which are close to the computer science area; three were assessed as principal, and one assessed as supplementary. The selected databases are the following:

- Scopus (Principal)
- ACM Digital Library (Principal)
- Science Direct (Principal)
- IEEE Xplore (Supplementary)

For this work, those will be treated simply as scientific databases.

2.4.2. Search Strings

Table 3 shows the search strings that were built based on the scope of the study, the PICOC table, and the research questions. Each search string is presented by the way it was used in each scientific database.

**Table 3.** Search strings.

| Search Engine | Search string | Additional Filter |
|---|---|---|
| Scopus | TITLE-ABS-KEY ( transpiler OR transcompiler OR "cross compiler" OR "cross-compiler" OR "source to source" OR "S2S compiler" OR "source-to-source" OR transpilation OR transcompilation OR "cross compilation" OR "S2S compilation" ) AND ( LIMIT-TO ( PUBYEAR , 2022 ) OR LIMIT-TO ( PUBYEAR , 2021 ) OR LIMIT-TO ( PUBYEAR , 2020 ) OR LIMIT-TO ( PUBYEAR , 2019 ) OR LIMIT-TO ( PUBYEAR , 2018 ) OR LIMIT-TO ( PUBYEAR , 2017 ) OR LIMIT-TO ( PUBYEAR , 2016 ) OR LIMIT-TO ( PUBYEAR , 2015 ) OR LIMIT-TO ( PUBYEAR , 2014 ) OR LIMIT-TO ( PUBYEAR , 2013 ) ) AND ( LIMIT-TO ( SUBJAREA , "COMP" ) ) | • Period: from 2013 to 2022 <br>• Area: Computer Science |
| IEEE Xplore | ("All Metadata":transpiler) OR ("All Metadata":transcompiler) OR ("All Metadata":"cross compiler") OR ("All Metadata":"cross-compiler") OR ("All Metadata":"source to source") OR ("All Metadata":"source-to-source") OR ("All Metadata":"S2S compiler") OR ("All Metadata":"transpilation") OR ("All Metadata":transcompilation) OR ("All Metadata":"cross compilation") OR ("All Metadata":"S2S compilation") | • Period: from 2013 to 2022 |
| ACM | Title:(transpiler OR transcompiler OR "cross compiler" OR "cross-compiler" OR "source to source" OR "S2S compiler" OR "source-to-source" OR transpilation OR transcompilation OR "cross compilation" OR "S2S compilation") OR Abstract:(transpiler OR transcompiler OR "cross compiler" OR "cross-compiler" OR "source to source" OR "S2S compiler" OR "source-to-source" OR transpilation OR transcompilation OR "cross compilation" OR "S2S compilation") OR Keywords:(transpiler OR transcompiler OR "cross compiler" OR "cross-compiler" OR "source to source" OR "S2S compiler" OR "source-to-source" OR transpilation OR transcompilation OR "cross compilation" OR "S2S compilation") | • Period: from 2013 to 2022 |
| Science Direct | "transpiler" OR "transcompiler" OR "cross compiler" OR "cross-compiler" OR "source to source" OR "S2S compiler" OR "source-to-source" OR "transpilation" OR "transcompilation" | • Period: from 2013 to 2022 <br>• Subject areas: computer science |

### 2.4.3. Inclusion and Exclusion Criteria

Table 4 defines the inclusion and exclusion criteria for this study.

**Table 4.** Inclusion and exclusion criteria.

| Criteria Type | Description | Definition |
|---|---|---|
| Period | Articles can be selected based on the time of the article publication date. | • Inclusion: Last 10 years, from 2013 to 2022.<br>• Exclusion: Before 2013 and also papers published after the search execution (Jan 2023) |
| Language | Articles can be excluded based on language. | • Exclusion: Articles published not in English. |
| Type of Literature | Articles can be excluded if they are considered grey literature | • Exclusion: Articles obtained from outside recognized scientific databases, newsletters, speeches, reports, non-published articles. |
| Document Type | Articles can be excluded by their document type | • Inclusion: Conference article, Conference Review, Journal Articles, Book chapters, and books.<br>• Exclusion: Short surveys, letters, notes, erratums. |
| Impact Source | Articles can be excluded by impact factor or quartile of the source. | No criteria will be applied |
| Accessibility | Articles can be excluded if they cannot be directly accessible or cannot obtain full-text version | • Exclusion: Articles for which the main tabulation data cannot be obtained, such as the title, the abstract, and the DOI code. Articles for which the full-text version cannot be obtained will also be excluded if this is necessary to advance to the next review step. |
| Relevance to research questions | Articles can be excluded if they cover topics not relevant to answer research questions. | • Exclusion: Articles that are not related to the research topic are excluded. |
| Duplicated articles | Articles can be excluded if they are repeated between scientific databases. | • Exclusion: Duplicate articles found in scientific databases are excluded. |

*2.5. Appraisal*

2.5.1. Selecting Studies

Queries were made in the four selected scientific databases, directly applying the filter criteria that search engines have. Of this, a total of 1181 articles were found, of which 663 were found in Scopus, 246 were found in IEEE Xplore, 186 were found in ACM, and 86 were found in Science Direct.

For the processing of the information obtained, the "SQL-based processing method" was used. The details of the aforementioned method are found in a specific section later in this article.

References were downloaded in BIB format from each database, and then this information was migrated to an SQL database using a custom migration program. A data

enrichment procedure, application of acceptance criteria, article tagging, and classification were carried out.

During the processing, it could be seen that a group of articles that did not have a DOI reference was marked. The reason is that these articles correspond to articles that were indexed in the scientific database, but they correspond to the cover or prologue of the journal or the presentation cover of the conference. After this filter, the number of articles was reduced to 1104.

Subsequently, there was identified some articles that were duplicated among the scientific databases. After this filter, the articles were reduced to 698 in number.

A quality review of the articles was carried out, validating that the proposed topic is related to transpilers and does not correspond to topics that could be confused with the search criteria. Finally, 683 articles remained, which is the base of articles considered for the data extraction and analysis stage. Figure 2 exposes the process of filtering until reaching the final list of articles to consider.

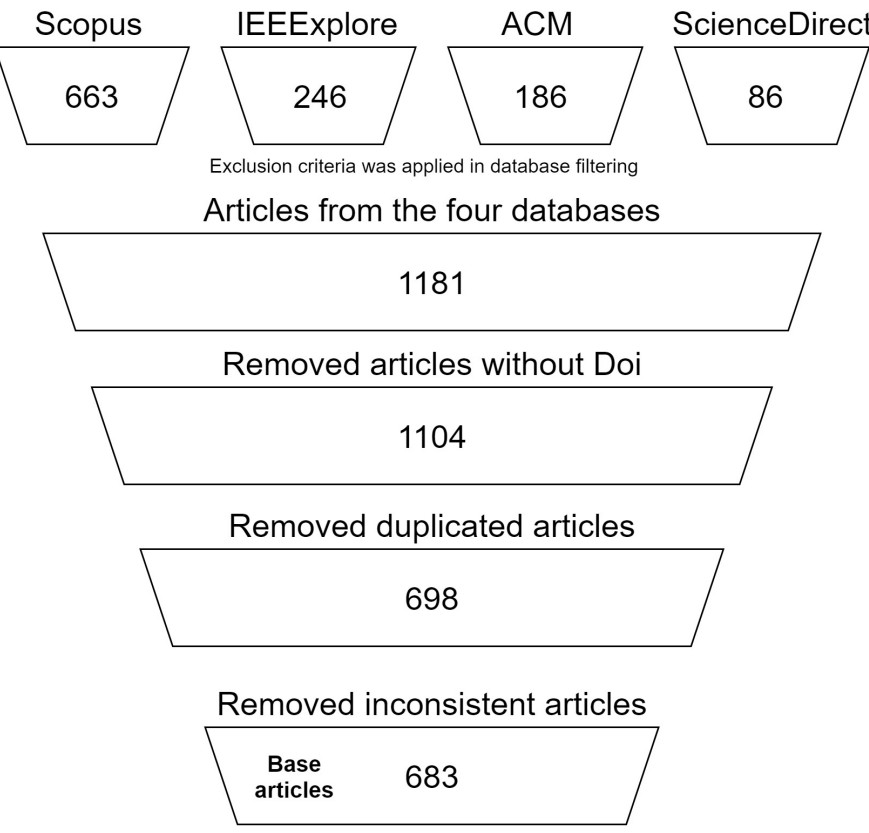

**Figure 2.** Article funnel.

### 2.5.2. Quality Assessments of Studies

Given that this is a broad search of articles for a root topic, it can be considered that there is a reasonably wide universe of articles available, so the quality criteria of the study will be focused on the efficiency of the exclusion and processing methods to prevent valid items from being wrongly excluded.

The application of the SQL-based processing method allows processing such as duplicate detection or automatic exclusion to not be a manual task that could introduce errors. The data enrichment process with alternative sources helps the downloaded information to be as complete and detailed as possible, which allows the selection of articles to be based

on the most reliable information possible. This process is explained in a separate section later in this document.

Manual review through quick readings of the chosen articles is a complementary practice that allows the chosen articles to have a very high level of representativeness. This is complemented by the verification of the presence of the keywords on the title or abstract of the chosen articles.

### 2.6. Synthesis

#### 2.6.1. Extract Data

The source information was obtained directly from scientific databases. A full extraction was carried out in a single moment. No source information update was considered later for this work. After the search criteria were applied in the search engine and the results were obtained, the information was exported in BibTeX format.

A utility was developed through a .Net Framework console program, importing the library called imbSCI (https://www.nuget.org/packages/imbSCI.BibTex/0.9.9 (accessed on 6 January 2023)), and also using the Entity Framework and the C# language. The objective of the program was to read the information from the files downloaded from the scientific databases and transform them into the SQL database data model.

Once the information was stored in the database, the developed console program was perfected to incorporate the consumption of the Crossref API (https://api.crossref.org/swagger-ui/index.html (accessed on 6 January 2023)) and the Crossref BibTeX service. Those APIs deliver the information of the consulted articles in JSON format and BibTeX, respectively, having a quantity of additional data that was updated directly in the SQL tables, mainly for the filling of previously empty information, or replacing fields with more quality information.

The data cleaning processes were also executed, marking the articles that at the end of the process did not have the minimum information filled out.

The result of the data extraction and preparation procedure consists of the availability of the information loaded in the tables of the SQL database, ready to start the respective processing and analysis.

#### 2.6.2. Categorize the Data

The data categorization process begins with the generation of the base terms list, in which each term is representative of a specific subarea of research. In this case, a general reading and a review of all the downloaded articles have been taken as a reference and a list has been generated based on the author's understanding. The topics were selected based on the understanding of certain research areas known to be related to the use of transpilers.

A word list was created. All the titles and the abstracts of the articles were extracted as separated plain texts and processed in a qualitative tabulation tool. Numbers, hyphens, and common words from the English language were removed. Once the word list was generated, it was exported to Excel. Table 5 shows the manual processing carried out with the data later, where it was incorporated with the "Language" mark when it is a non-representative language element, "Generic" when it is a term that does not determine a particular topic, and "Base List" when it is a representative element.

A word cloud was created, the objective being a visual representation of the words that have the highest recurrence among the base articles of the study. Figure 3 shows the cloud of words generated using the qualitative tabulation tool. In this, it is possible to see agreement with the word list, so several additional terms were chosen to add to the list of classification terms.

**Table 5.** Terms extracted from the word list processing

| Word | Length | Matches |
|---|---|---|
| performance | 11 | 541 |
| source-to-source | 16 | 471 |
| compiler | 8 | 450 |
| programming | 11 | 399 |
| parallel | 8 | 307 |
| software | 8 | 278 |
| memory | 6 | 271 |
| framework | 9 | 258 |
| transformation | 14 | 254 |
| hardware | 8 | 234 |
| optimization | 12 | 215 |
| development | 11 | 214 |
| gpu | 3 | 162 |
| algorithm | 9 | 156 |
| quantum | 7 | 138 |
| opencl | 6 | 134 |
| openmp | 6 | 133 |
| compilers | 9 | 123 |
| hls | 3 | 118 |
| architecture | 12 | 115 |
| gpus | 4 | 108 |
| javascript | 10 | 107 |
| kernels | 7 | 104 |
| simulation | 10 | 97 |
| fpga | 4 | 94 |
| kernel | 6 | 91 |
| semantics | 9 | 91 |
| cuda | 4 | 82 |
| learning | 8 | 82 |
| accelerators | 12 | 80 |
| benchmarks | 10 | 78 |
| mobile | 6 | 78 |
| power | 5 | 76 |
| network | 7 | 71 |
| space | 5 | 67 |
| benchmark | 9 | 66 |
| java | 4 | 66 |
| web | 3 | 66 |
| fpgas | 5 | 63 |
| graph | 5 | 62 |
| hpc | 3 | 62 |
| openacc | 7 | 61 |
| python | 6 | 61 |
| cloud | 5 | 59 |
| matlab | 6 | 59 |
| intel | 5 | 58 |
| polyhedral | 10 | 55 |
| low-level | 9 | 54 |
| portability | 11 | 54 |
| accelerator | 11 | 52 |
| translator | 10 | 51 |

Terms commonly used to define methods, techniques, platforms, or technologies were incorporated. In this case, these were mainly relationships with different operating systems, specific types of hardware or software, and elements of hardware and software architecture, among others.

**Figure 3.** Word cloud

Cleaning and ordering were carried out to choose the terms that could cause difficulties in their search, due to their easy coincidence or their short length.

Figure 4 shows the final list of tags used for data classification. The relationship procedure was executed through SQL queries. A temporary table was created with all the terms and a textual search was made in the title and abstract of each article, looking for the determined terms. In case of a match, a record in the relationship table (EntryTag) is created to indicate that relationship with the term. An article could have many relationships with the terms executed since the mixture and co-occurrence of terms in the same article will allow commonly combined themes to be determined.

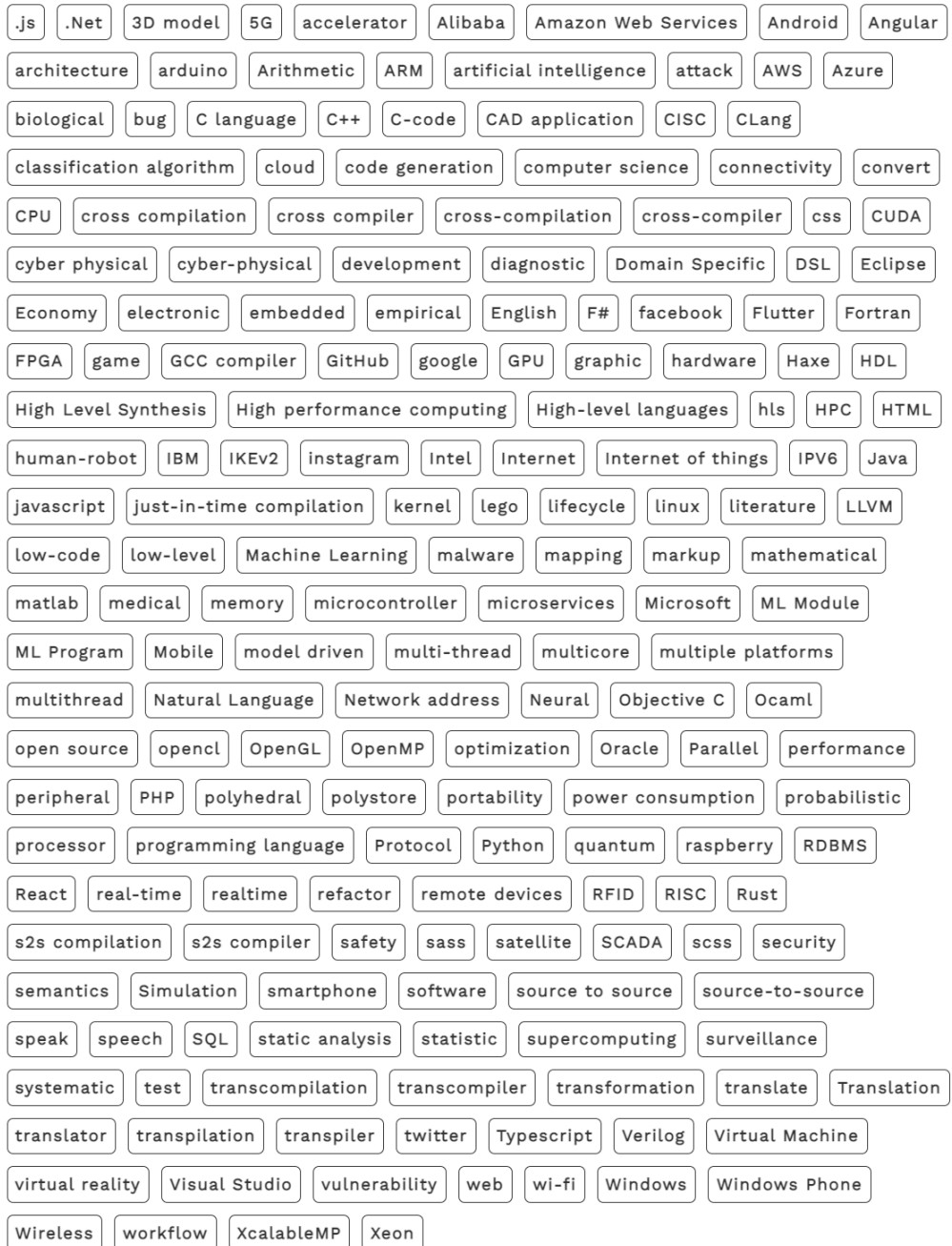

**Figure 4.** Considered classification tags

## 3. Results

The process of automatically matching the tags with the articles was executed, making a complete textual match in the title and abstract fields. With the list of tags proposed, it was possible to associate 98.975% of the articles with at least one tag. The 96.93% have more than one tag associated with them.

Below is the list of tags with the number of articles in which they match:
source-to-source (386), performance (305), transformation (231), Parallel (201), architecture (156), optimization (155), software (152), hardware (137), development (133), memory (128), GPU (115), test (103), Translation (90), kernel (83), programming language (81), processor (79), C++ (73), translate (72), accelerator (70), embedded (70), CPU (65), Java (63), Simulation (62), semantics (61), FPGA (58), Intel (58), web (48), C language (46), OpenMP (45), opencl (44), bug (42), code generation (40), mapping (39), translator (38), CUDA (38), javascript (38), transpiler (37), portability (37), hls (36), convert (35), multicore

(35), Mobile (34), security (32), cross-compilation (32), cloud (32), graphic (32), ARM (32), polyhedral (32), HPC (31), low-level (31), source to source (30), DSL (25), literature (24), transpilation (24), safety (23), cross-compiler (23), Rust (22), open source (21), Python (21), Machine Learning (21), Arithmetic (20), Fortran (20), Android (20), real-time (20), quantum (20), linux (19), matlab (18), LLVM (18), IBM (17), CLang (17), static analysis (16), power consumption (15), Neural (15), systematic (15), High performance computing (15), multi-thread (13), empirical (13), Xeon (13), Protocol (13), refactor (13), Virtual Machine (12), workflow (12), mathematical (11), transcompiler (11), HTML (10), Internet (10), smartphone (10), electronic (10), microcontroller (9), medical (9), Windows (9), HDL (9), Domain Specific (9), connectivity (9), statistic (8), google (8), Wireless (8), High Level Synthesis (8), computer science (8), attack (7), multithread (7), Angular (7), cross compiler (7), vulnerability (7), game (7), cross compilation (6), Internet of things (6), Microsoft (6), High-level languages (6), GitHub (6), SQL (6), probabilistic (5), React (5), English (5), multiple platforms (4), artificial intelligence (4), Verilog (4), Windows Phone (3), RISC (3), satellite (3), just-in-time compilation (3), .js (3), supercomputing (3), AWS (3), peripheral (3), raspberry (3), Oracle (2), C-code (2), surveillance (2), Ocaml (2), biological (2), PHP (2), transcompilation (2), css (2), facebook (2), malware (2), Economy (2), lego (2), Amazon Web Services (2), Typescript (2), Haxe (2), Azure (2), Natural Language (2), XcalableMP (1), human-robot (1), Eclipse (1), 5G (1), arduino (1), twitter (1), low-code (1), GCC compiler (1), SCADA (1), speech (1), .Net (1), classification algorithm (1), Objective C (1), lifecycle (1), speak (1), cyber-physical (1), realtime (1), microservices (1)

Most of the articles chosen for the study came from the Scopus scientific database. In Figure 5 it is possible to see the distribution of the articles according to their scientific database origin. It is noteworthy that some articles were duplicated among the different scientific databases, which for this graph have already been withdrawn.

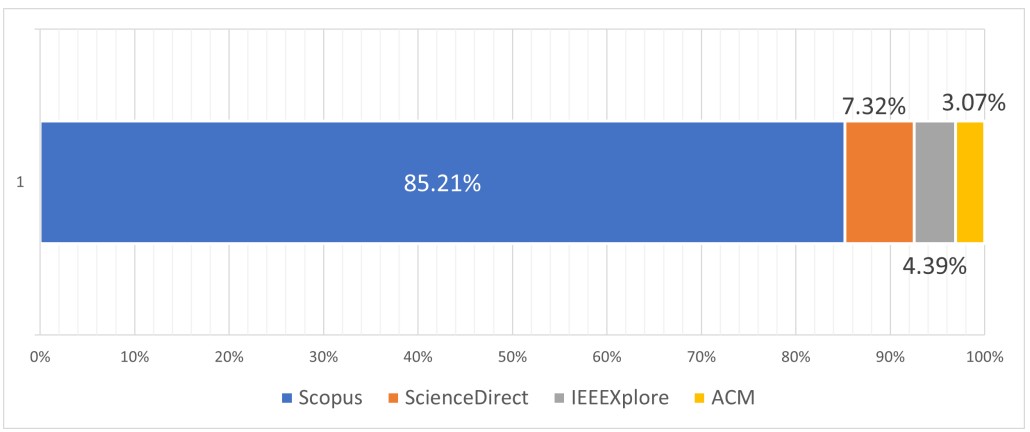

**Figure 5.** Articles distribution by scientific database.

The trend of growth or decrease of articles published on the subject over the years determines the level of interest that is maintained in the main topic. In Figure 6 it is possible to see the trend of the articles separated by scientific database, and also the global trend, which represents the sum of the others.

In a linear regression process on the global trend line, Equation (1) presents the formula in which it is possible to see that the number of articles has a slight negative trend; however, its representation explains almost a stable trend of publication on this topic. Therefore, it is expected that for the following years, this trend will continue.

$$y = -0.2121x + 69.467 \qquad (1)$$

From the selected articles, you can see those with the highest number of references. These articles can be considered representative elements of the detailed analysis since they have had a high impact on the world of research. In Table 6 it is possible to see the

15 most-referenced articles within the articles chosen for the present study. It should be noted that the citation data corresponds to the update obtained through the Crossref API, which does not always have an immediate synchronization of the citations, but has a near reference.

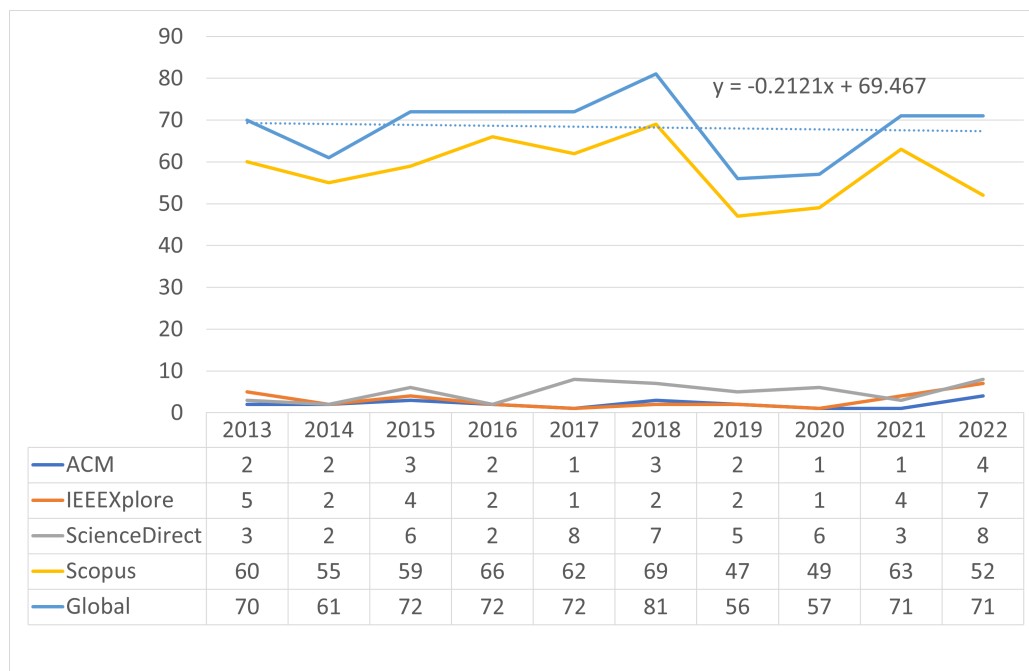

**Figure 6.** Trends of articles between years.

Another key indicator to identify the articles with the greatest relevance and impact in the research is the identification of the number of references among the articles selected for this study. In Table 7 it is possible to see the 10 most-referenced articles among them.

Due to the different review procedures of articles published in journals, book chapters, and articles published in conferences, it is important to review the proportion of these, to determine the type of article commonly chosen by researchers for publication in terms of the transpilers. In Figure 7 you can see an almost similar ratio between journal articles and conference articles.

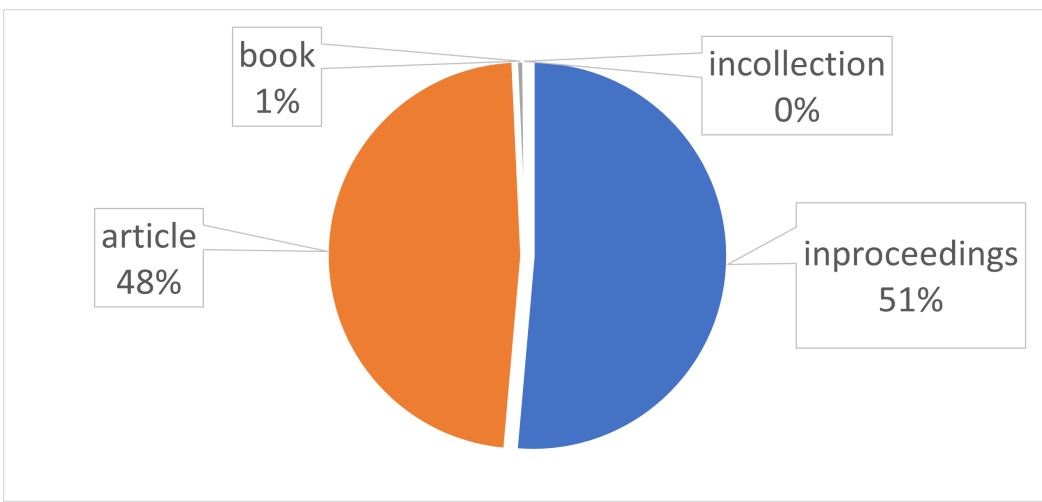

**Figure 7.** Articles per type.

**Table 6.** Top 15 most-referenced articles

| Article | # References |
|---|---|
| Automated Systolic Array Architecture Synthesis for High Throughput CNN Inference on FPGAs [27] | 210 |
| Polyhedral parallel code generation for CUDA [28] | 203 |
| Consumer perceptions of information helpfulness and determinants of purchase intention in online consumer reviews of services [29] | 166 |
| Checking and Enforcing Robustness against TSO [30] | 75 |
| Design and evaluation of gradual typing for python [31] | 67 |
| High-level synthesis of dynamic data structures: A case study using Vivado HLS [32] | 66 |
| HIPA: A Domain-Specific Language and Compiler for Image Processing [33] | 65 |
| Smells in software test code: A survey of knowledge in industry and academia [34] | 64 |
| Robust optimization to secure urban bulk water supply against extreme drought and uncertain climate change [35] | 64 |
| A large-scale cross-architecture evaluation of thread-coarsening [36] | 59 |
| PORPLE: An Extensible Optimizer for Portable Data Placement on GPU [37] | 58 |
| An automatic input-sensitive approach for heterogeneous task partitioning [38] | 56 |
| Weighted Superposition Attraction (WSA): A swarm intelligence algorithm for optimization problems—Part 1: Unconstrained optimization [39] | 50 |
| Towards a Compiler for Reals [40] | 50 |
| Opening the Duke electronic health record to apps: Implementing SMART on FHIR [41] | 49 |

**Table 7.** Top 10 most-referenced among the selected articles.

| Article | # References |
|---|---|
| Performance-driven instrumentation and mapping strategies using the LARA aspect-oriented programming approach [42] | 10 |
| HIPA: A Domain-Specific Language and Compiler for Image Processing [33] | 8 |
| The Cetus Source-to-Source Compiler Infrastructure: Overview and Evaluation [43] | 5 |
| Clava: C/C source-to-source compilation using LARA [44] | 5 |
| Cross-Compiling Android Applications to iOS and Windows Phone 7 [45] | 4 |
| Parallel tiled Nussinov RNA folding loop nest generated using both dependence graph transitive closure and loop skewing [46] | 4 |
| TRACO Parallelizing Compiler [47] | 3 |
| LARA as a language-independent aspect-oriented programming approach [48] | 3 |
| Polyhedral Bubble Insertion: A Method to Improve Nested Loop Pipelining for High-Level Synthesis [49] | 3 |
| Source-to-Source Parallelization Compilers for Scientific Shared-Memory Multi-core and Accelerated Multiprocessing: Analysis, Pitfalls, Enhancement, and Potential [50] | 3 |

In the case of articles that have been published in journals, it is important to identify which are the most important journals that researchers have used for the publication of issues related to transpilers. In Table 8 it is possible to see the list of the 10 most used journals by researchers on this topic. The quartile information was taken from Scimago Jr. website (https://www.scimagojr.com/ (accessed on 6 January 2023)), considering the software subject specifically.

**Table 8.** Top 10 most-used journals.

| Journal | Article Number | Quartile |
| :---: | :---: | :---: |
| Proceedings of the ACM on Programming Languages (https://dl.acm.org/journal/pacmpl (accessed on 10 January 2023)) | 9 | Q1 |
| The Journal of Supercomputing (https://www.springer.com/journal/11227 (accessed on 10 January 2023)) | 9 | Q2 |
| Advances in Intelligent Systems and Computing (https://www.springer.com/series/11156 (accessed on 10 January 2023)) | 9 | Q4 |
| Communications in Computer and Information Science (https://www.springer.com/series/7899 (accessed on 10 January 2023)) | 9 | Q4 |
| ACM Transactions on Architecture and Code Optimization (https://dl.acm.org/journal/taco (accessed on 10 January 2023)) | 8 | Q3 |
| International Journal of Parallel Programming (http://www.springer.com/computer+science/ theoretical+computer+science/foundations+of+ computations/journal/10766 (accessed on 10 January 2023)) | 8 | Q3 |
| IEEE Transactions on Computer-Aided Design of Integrated Circuits and Systems (https://ieeexplore.ieee.org/xpl/RecentIssue. jsp?punumber=43 (accessed on 10 January 2023)) | 7 | Q2 |
| Parallel Computing (https://www.journals. elsevier.com/parallel-computing (accessed on 10 January 2023)) | 7 | Q3 |
| Concurrency and Computation: Practice and Experience (https: //onlinelibrary.wiley.com/journal/15320634 (accessed on 10 January 2023)) | 6 | Q3 |
| Journal of Parallel and Distributed Computing (https://www.sciencedirect.com/journal/ journal-of-parallel-and-distributed-computing (accessed on 10 January 2023)) | 5 | Q1 |

As part of the information obtained in data enrichment, it is possible to have a list of subjects that each scientific journal proposes as its main focus. By tabulating all the related journal data, it has been possible to visualize which are the most important subjects of the journals in the context of the transpilers and the selected articles. Figure 8 shows a treemap in which the most recurrent subjects are presented larger.

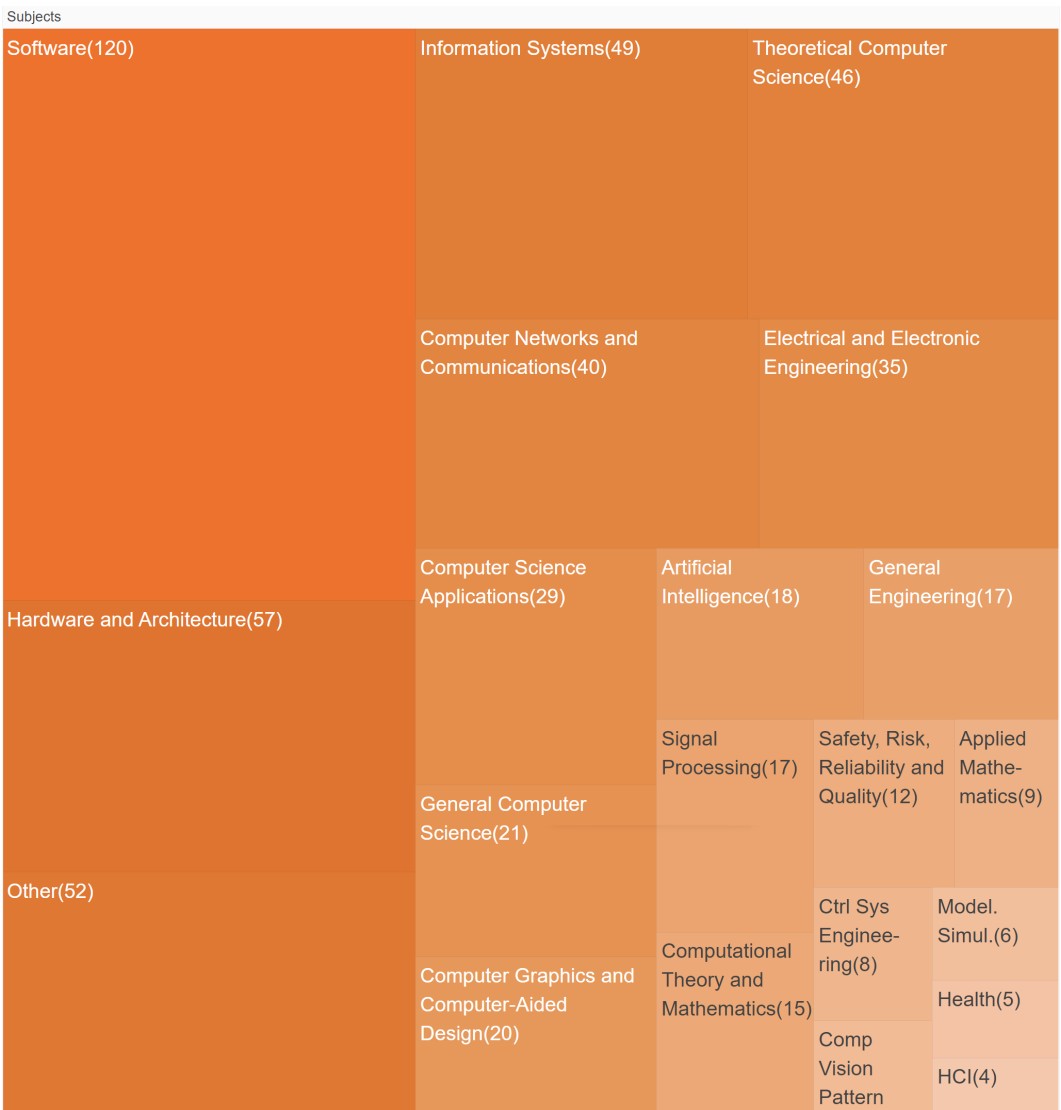

**Figure 8.** Journal subjects tree map.

In the case of papers published at conferences, it is also important to identify which conferences are the most widely used by researchers for their transpiler-related publications. In Table 9 it is possible to see the list of the 10 conferences most used by researchers.

Analyzing the authors of the most recurring publications among the selected articles makes it possible to identify the main ones in the use of transpilers and their research areas. In Table 10 it is possible to see the list of the 10 most recurring authors who appear in the first position in their articles. In many journals and conferences, it is understood that the first author is listed as the main one. Additionally, in Table 11 you can see the list of most recurring authors regardless of the position in which they are in their publications.

There are certain articles selected for this study which have a significant amount of relationship with the proposed tags. This serves to highlight certain articles that address central themes in reference to the proposed tags. Therefore, it is important to consider reading them in detail, since they could present basic information on the use of transpilers. Table 12 presents the top ten articles which have the most relationship tags.

A tag co-occurrence analysis was performed on the analyzed articles. The objective is to find the tags that in combination appear similar within several articles. This will allow for determining which combinations of concepts are the most common among the selected articles. In Table 13 it is possible to see the first 10 most repeated tag combinations. For this analysis, terms synonymous with transpiler were removed.

**Table 9.** Top 10 most-used conferences.

| Acronym | Conference Name | Article Number |
|---|---|---|
| CGO | IEEE/ACM International Symposium on Code Generation and Optimization (CGO) | 10 |
| PLDI | ACM SIGPLAN International Conference on Programming Language Design and Implementation | 10 |
| SPLASH | SPLASH : Conference on Systems, Programming, and Applications: Software for Humanity | 9 |
| SC | SC: The International Conference for High Performance Computing, Networking, Storage and Analysis | 7 |
| PPoPP | PPoPP: ACM SIGPLAN Symposium on Principles and Practice of Parallel Programming | 7 |
| IPDPSW | IEEE International Symposium on Parallel & Distributed Processing, Workshops and Phd Forum (IPDPSW) | 7 |
| ICS | ACM international conference | 7 |
| ICSE | ICSE: International Conference on Software Engineering | 6 |
| FCCM | International Symposium on Field-Programmable Custom Computing Machines (FCCM) | 6 |
| FedCSIS | Conference on Computer Science and Intelligence Systems | 5 |

**Table 10.** Top 10 most-recurrent authors (first author).

| Author | # Articles |
|---|---|
| Marek Palkowski | 9 |
| Alejandro Acosta | 6 |
| João Bispo | 6 |
| Aurelien Bloch | 5 |
| Pedro Pinto | 4 |
| Junyi Liu | 4 |
| Bo Qiao | 4 |
| Patryk Chaber | 4 |
| Arno Puder | 4 |
| Cedric Nugteren | 3 |

**Table 11.** Top 10 most-recurrent authors (any position).

| Author | # Articles |
|---|---|
| Joao Manuel Paiva Cardoso | 21 |
| João Bispo | 16 |
| Marek Palkowski | 12 |
| Wlodzimierz Bielecki | 11 |
| George A. Constantinides | 9 |
| Frank Hannig | 9 |
| Jurgen Teich | 9 |
| Steven Derrien | 8 |

**Table 11.** *Cont.*

| Author | # Articles |
|---|---|
| Alejandro Acosta | 8 |
| Pedro Pinto | 7 |

**Table 12.** Top 10 articles with more tag coincidence.

| Article Title | Matches | Tags |
|---|---|---|
| OP2-Clang: A Source-to-Source Translator Using Clang/LLVM LibTooling [51] | 28 | performance, HPC, optimization, OpenMP, hardware, GPU, CLang, Parallel, Translation, C language, translator, embedded, Fortran, CPU, Domain Specific, architecture, processor, portability, source-to-source, transformation, translate, memory, Intel, LLVM, C++, DSL, refactor, CUDA |
| Source-to-source translation: Impact on the performance of high level synthesis [52] | 23 | statistic, performance, Translation, hardware, source-to-source, memory, architecture, High performance computing, High Level Synthesis, CUDA, Verilog, OpenMP, software, Intel, High-level languages, optimization, HDL, hls, C-code, Microsoft, CPU, C++, FPGA |
| A source-to-source CUDA to SYCL code migration tool: Intel DPC++ Compatibility Tool [53] | 17 | accelerator, CPU, software, hardware, C++, LLVM, source-to-source, architecture, Intel, CLang, performance, Parallel, GPU, FPGA, CUDA, multiple platforms, development |
| Paper: Togpu: Automatic Source Transformation from C++ to CUDA using Clang/LLVM [54] | 17 | transformation, software, C++, Translation, open source, translate, CUDA, GPU, development, LLVM, optimization, CLang, workflow, Parallel, opencl, performance, source to source |
| Automatic OpenCL Code Generation from LLVM-IR using Polyhedral Optimization [55] | 17 | code generation, CPU, polyhedral, LLVM, CUDA, C-code, C++, kernel, CLang, architecture, transformation, opencl, optimization, FPGA, source-to-source, Parallel, GPU |
| Transpiler-based architecture for multi-platform web applications [16] | 17 | translate, GitHub, Java, architecture, transpilation, programming language, Haxe, cloud, test, performance, css, web, software, transpiler, PHP, HTML, javascript |
| Pocket code [56] | 16 | Windows Phone, Parallel, C++, Windows, Java, open source, HTML, programming language, development, test, Mobile, javascript, smartphone, Android, cross-compilation, lego |
| Modernizing the NEURON Simulator for Sustainability, Portability, and Performance [57] | 16 | C language, architecture, Simulation, source-to-source, development, software, GPU, portability, performance, hardware, React, biological, cloud, just-in-time compilation, memory, workflow |
| Translating CUDA to OpenCL for Hardware Generation using Neural Machine Translation [58] | 16 | source-to-source, Translation, C language, DSL, opencl, kernel, hls, High-level languages, hardware, CUDA, FPGA, GPU, software, C++, translate, Neural |
| Automatic Kernel Fusion for Image Processing DSLs [59] | 15 | performance, optimization, source-to-source, memory, software, kernel, open source, portability, GPU, Parallel, DSL, graphic, hardware, accelerator, C language |

**Table 13.** Co-ocurrence analysis.

| Tags Combination | Number |
|---|---|
| (performance) (Parallel), | 127 |
| (transformation) (performance), | 105 |
| (performance) (optimization), | 97 |
| (performance) (architecture), | 95 |
| (performance) (hardware), | 87 |
| (performance) (GPU), | 83 |
| (performance) (memory), | 79 |
| (transformation) (Parallel), | 79 |
| (transformation) (optimization), | 75 |
| (Parallel) (architecture), | 72 |
| (Parallel) (GPU), | 68 |
| (Parallel) (optimization), | 64 |
| (software) (performance), | 63 |
| (Parallel) (memory), | 62 |
| (hardware) (architecture), | 59 |
| (software) (hardware), | 58 |
| (performance) (development), | 57 |
| (transformation) (software), | 57 |
| (transformation) (memory), | 52 |
| (software) (development), | 50 |

For the articles with the highest frequency of association, a complimentary review was carried out, identifying the most recurring global concepts in the use of transpilers. From them, it was possible to identify the classification terms, which group several tags and which represent the uniqueness of a research area.

This was performed by reading the abstract and the title of the most frequent tags' articles and doing a list of second-level tags. Then, concept existence testing was carried out by comparing with the main related articles to confirm that there is a group of works with a high relation or also a medium relation between them. Despite some articles perhaps using a term for other interpretations, it was assumed that the majority of them will relate to the same main concepts.

Table 14 shows the main concepts found, the list of related tags to the concept, the number of articles, the number of references to those articles, and a small list of example articles that correspond to this concept. This table was created by running the code shown in Figure 9.

It should be noted that an article could be related to several tags and several concepts at the same time, increasing the sum by more than one concept. The sum of all the considered articles for this work will determine which are the most recurring research areas for the use of transpilers.

**Table 14.** Comprehensive mapping chart.

| Concept | Tags | # Articles | # References | Examples |
|---|---|---|---|---|
| transpiler | cross compilation, cross compiler, cross-compilation, cross-compiler, hls, source to source, source-to-source, source-to-source compilation, source-to-source compiler, transcompilation, transcompiler, transformation, transpilation, transpiler | 563 | 3262 | [51,60,61] |
| performance | High performance computing, HPC, performance, supercomputing | 310 | 2345 | [53,62,63] |
| parallel | CUDA, GPU, kernel, opencl, OpenMP, optimization, Parallel, XcalableMP | 348 | 2212 | [64–66] |
| development | architecture, development, Eclipse, GitHub, lifecycle, open source, portability, software | 351 | 1965 | [67–69] |
| programming language | .js, .Net, C language, C++, C-code, CLang, F#, Fortran, Haxe, High-level languages, Java, javascript, low-code, markup, matlab, Objective C, Ocaml, PHP, programming language, Python, Rust, semantics | 308 | 1667 | [70–72] |
| hardware | CISC, CPU, electronic, hardware, HDL, Intel, microcontroller, peripheral, Verilog | 235 | 1472 | [73–75] |
| compiler | GCC compiler, just-in-time compilation, LLVM, low-level, memory, polyhedral | 179 | 1427 | [47,76,77] |
| testing | bug, diagnostic, portability, static analysis, test | 174 | 1201 | [78–80] |
| graphics | 3D model, CAD application, GPU, graphic, OpenGL | 128 | 1091 | [81–83] |
| embedded systems | arduino, cyber physical, cyber-physical, embedded, FPGA, Internet of things, microcontroller, raspberry, real-time, realtime, remote devices | 145 | 1046 | [52,84,85] |
| processor | ARM, CISC, CPU, Intel, multi-thread, multicore, multithread, processor, RISC, Xeon | 214 | 1030 | [52,86,87] |
| language | English, semantics, speak, speech, translate, Translation, translator | 193 | 998 | [88–90] |
| other | accelerator, computer science, convert, High Level Synthesis, quantum | 135 | 777 | [59,91,92] |
| operating system | kernel, linux, virtual reality, Windows | 97 | 683 | [93–95] |
| code generation | code generation, Domain Specific, DSL, model driven, refactor | 75 | 662 | [96–98] |
| architecture | microservices, multi-thread, multiple platforms, Simulation | 78 | 532 | [99–101] |
| security | attack, IKEv2, malware, safety, security, surveillance, vulnerability | 56 | 451 | [102–104] |
| research | empirical, literature, mapping, systematic | 84 | 438 | [76,105,106] |
| AI | artificial intelligence, classification algorithm, human-robot, Machine Learning, ML Module, ML Program, Natural Language, Neural | 39 | 426 | [107–109] |
| math | Arithmetic, mathematical, probabilistic, statistic | 43 | 347 | [52,110,111] |
| front-end | .js, Angular, css, Flutter, HTML, javascript, React, sass, Typescript, web | 72 | 321 | [112–114] |

**Table 14.** *Cont.*

| Concept | Tags | # Articles | # References | Examples |
|---|---|---|---|---|
| mobile | Android, Mobile, smartphone, Windows Phone | 40 | 273 | [115–117] |
| networking | 5G, connectivity, Internet, IPV6, Protocol, satellite, Virtual Machine, wi-fi, Wireless | 53 | 188 | [118–120] |
| cloud | Amazon Web Services, AWS, Azure, cloud | 32 | 123 | [121–123] |
| brand | Alibaba, Amazon Web Services, AWS, Azure, google, IBM, Microsoft, Oracle | 30 | 111 | [52,124,125] |
| energy | power consumption | 15 | 83 | [126–128] |
| quantum | quantum | 20 | 55 | [105,129,130] |
| vertical | biological, Economy, medical | 13 | 38 | [57,131,132] |
| games | game, lego | 9 | 36 | [56,133,134] |
| industry | cyber physical, cyber-physical, RFID, SCADA, workflow | 14 | 28 | [135–137] |
| database | Oracle, polystore, RDBMS, SQL | 8 | 27 | [138–140] |
| social network | facebook, twitter | 3 | 24 | [141–143] |

```sql
select w.taggroup TagGroup,
    (select string_agg(t.TAG, ', ')
        from taggroup t
        where t.taggroup = w.taggroup) as tags,
    (select count(b1.EntryID)
        from bibentry b1
        where b1.EntryStatus = 1 and b1.entryid in
        (select t1.entryid
            from entrytag t1 inner join
                taggroup g1 on t1.TAG = g1.tag
        where g1.taggroup = w.taggroup)) ArticleNumber,
    (select SUM(b1.isreferencedbycount)
        from bibentry b1
        where b1.EntryStatus = 1 and b1.entryid in
        (select t1.entryid
            from entrytag t1 inner join
                taggroup g1 on t1.TAG = g1.tag
            where g1.taggroup = w.taggroup)) ReferenceNumber,
    (select string_agg('\cite{' + ww.EntryKey + '}', '')
    from
        (select row_number() over (order by newid() desc) as num,
        b1.doi, b1.EntryKey
        from bibentry b1
        where b1.EntryStatus=1 and b1.entryid in
        (select t1.entryid
            from entrytag t1 inner join
                taggroup g1 on t1.TAG = g1.tag
            where g1.taggroup = w.taggroup)) as ww
    where ww.num <=3) as Dois
from (
    select g.taggroup, count(*) as cuenta
    from taggroup g
    group by g.taggroup
) as w
order by 4 desc
```

**Figure 9.** SQL code for the Comprehensive mapping chart

Since transpilers involve conversion between programming languages, it is important to analyze which programming languages are mainly involved in these transformation processes. In Table 15, it is possible to see the most-referenced programming languages among the selected articles.

**Table 15.** Top 10 most recurrent programming languages.

| Programming Language | # Articles |
|:---:|:---:|
| C++ / C | 135 |
| Java | 63 |
| javascript | 41 |
| Rust | 22 |
| Python | 21 |
| Fortran | 20 |
| matlab | 18 |
| High-level languages | 6 |
| Haxe | 2 |
| Ocaml | 2 |

As indicated in the disambiguation section of this study, transpilers have multiple synonyms by which the literature refers to them. In Table 16, it is possible to see the most-used synonym within the framework of the selected studies.

**Table 16.** Most-used transpiler synonyms.

| Term | # Articles |
|:---:|:---:|
| source-to-source | 386 |
| transformation | 231 |
| transpiler | 37 |
| hls | 36 |
| cross-compilation | 32 |
| source to source | 30 |
| transpilation | 24 |
| cross-compiler | 23 |
| transcompiler | 11 |
| cross compiler | 7 |
| cross compilation | 6 |
| transcompilation | 2 |

## 4. Industry Applications

A search was conducted using Google Search and referring to technical forums, searching about transpilers, their synonyms, and another kind of source code transformations. In addition, we considered their usage in applications or frameworks. Then was performed via deep navigation and followed by references to find additional related techniques. The following main references have been found:

- Haxe cross-compiler, which can transform Haxe language to several target languages (https://haxe.org/ (accessed on 11 January 2023))

- Typescript, which can transform the typescript language to javascript (https://www.typescriptlang.org/ (accessed on 11 January 2023))
- Dart is a client-optimized programming language which can compile for several platforms such as Android, iOS, Windows, and linux, and can also generate the code for Javascript. (https://dart.dev/overview (accessed on 11 January 2023))
- React Native is a cross-platform framework that uses the javascript language as a base, and can generate native code for several platforms such as Android, iOS, and the web. (https://reactnative.dev/ (accessed on 11 January 2023))
- Angular is a web front-end framework based on Typescript to build small to enterprise-level applications. (https://angular.io/ (accessed on 11 January 2023))
- The .Net framework proposes the common intermediate language (CIL), which is a human-readable code. When compatible programming language such as C#, Visual Basic .Net, F#, and others perform the compiling process, they really are performing a transpilation process to CIL. (https://dev.to/kcrnac/net-execution-process-explained-c-1b7a (accessed on 11 January 2023))
- Sharpkit is a tool to transform C# to Javascript (https://sharpkit.github.io/ (accessed on 11 January 2023))
- JSIL is a tool to transform CIL to Javascript (http://jsil.org/ (accessed on 11 January 2023))
- JSSweet is a tool that transpiles Java to Javascript (https://www.jsweet.org/ (accessed on 11 January 2023))
- Bck2Brwsr is a tool capable to transform Java Bytecode back to Javascript (http://wiki.apidesign.org/wiki/Bck2Brwsr (accessed on 11 January 2023))
- Cross-compilation for image preparation for Docker (https://www.docker.com/blog/faster-multi-platform-builds-dockerfile-cross-compilation-guide/ (accessed on 11 January 2023))
- Cross-compilation using the Go programming language (https://opensource.com/article/21/1/go-cross-compiling (accessed on 11 January 2023))
- FizzedWL is a tool to transform C# to Haxe (https://github.com/FizzerWL/Cs2hx (accessed on 11 January 2023))
- CoffeeScript is a language that transpile to javascript (https://coffeescript.org/ (accessed on 11 January 2023))
- SASS is a language that can transpile to CSS (https://sass-lang.com/ (accessed on 11 January 2023))
- The WORA definition (write-once run anywhere) proposed by Sun Microsystems (https://www.techtarget.com/whatis/definition/write-once-run-anywhere-WORA (accessed on 11 January 2023))
- Facebook AI Research' Transcoder (https://arxiv.org/pdf/2006.03511.pdf (accessed on 11 January 2023))

## 5. The SQL-Based Processing Method

The objective of defining a new data processing method for literature review and literature mapping is to expedite the processing of the large amount of data that can be generated in this type of study, especially if the evaluations are carried out with the premise of carrying out an investigation of a root topic.

The proposed method has three main phases: (a) data collection, (b) data processing, and (c) result reporting. Each phase has several internal steps, as shown in Figure 10. The breakdown of the operation of each of the steps of the method is presented below.

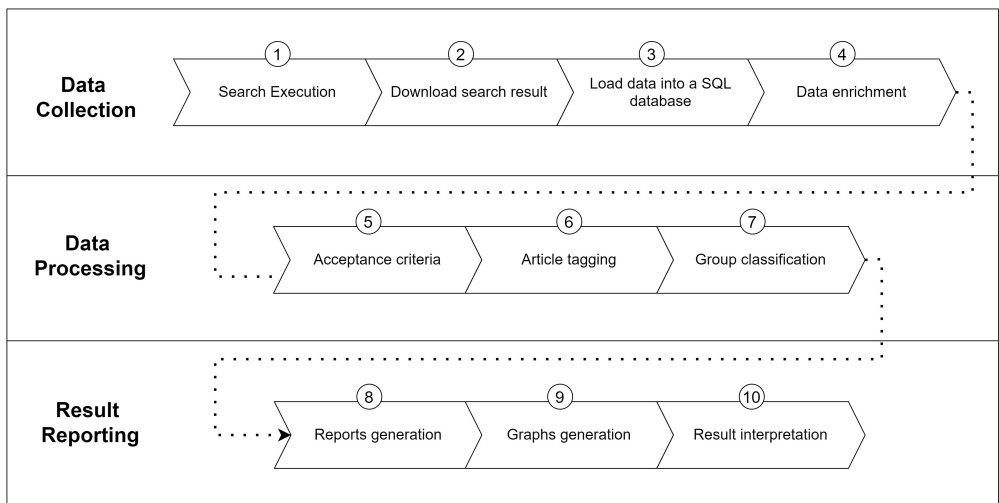

**Figure 10.** Steps of the processing method.

### 5.1. Search Execution

After the study planning stage, the search strings are run directly against the chosen scientific databases. In many cases, this consists of using operators or criteria applied through the advanced search or using complementary filters. The method starts once the database searches have been performed.

### 5.2. Download Search Result (BibTeX Format)

Scientific databases provide massive export mechanisms for the data resulting from the query made containing all the matching records. The processing method requires extracting the complete data in BibTeX format. If the database allows it, all the metadata that is allowed should be incorporated, in order to have the greatest amount of information possible.

In the case of Scopus, IEEE Xplore, and ACM, it is possible to find different ways to export the BibTeX format, although the amount of data that is filled in the format differs between them.

### 5.3. Load Data into a SQL Database

A database has been designed that has the structure presented in Figure 11. The main table is called "BibEntry", in which the records of the Doi files downloaded are recorded one by one. Some tables are referred to or filled later.

Regarding the transformation of data from the BibTeX format to the relational model, a console program that allows direct conversion is used (development carried out as part of this research). This program was developed using the .Net Framework, Entity Framework (https://learn.microsoft.com/es-es/ef/ef6/ (accessed on 22 December 2022)), and the imbSCI library through Nuget (https://www.nuget.org/packages/imbSCI.BibTex/0.9.9 (accessed on 22 December 2022)).

The procedure consists of reading the bib file and using its detailed information to fill in the database tables. For each loading process, the origin of each record has to be identified.

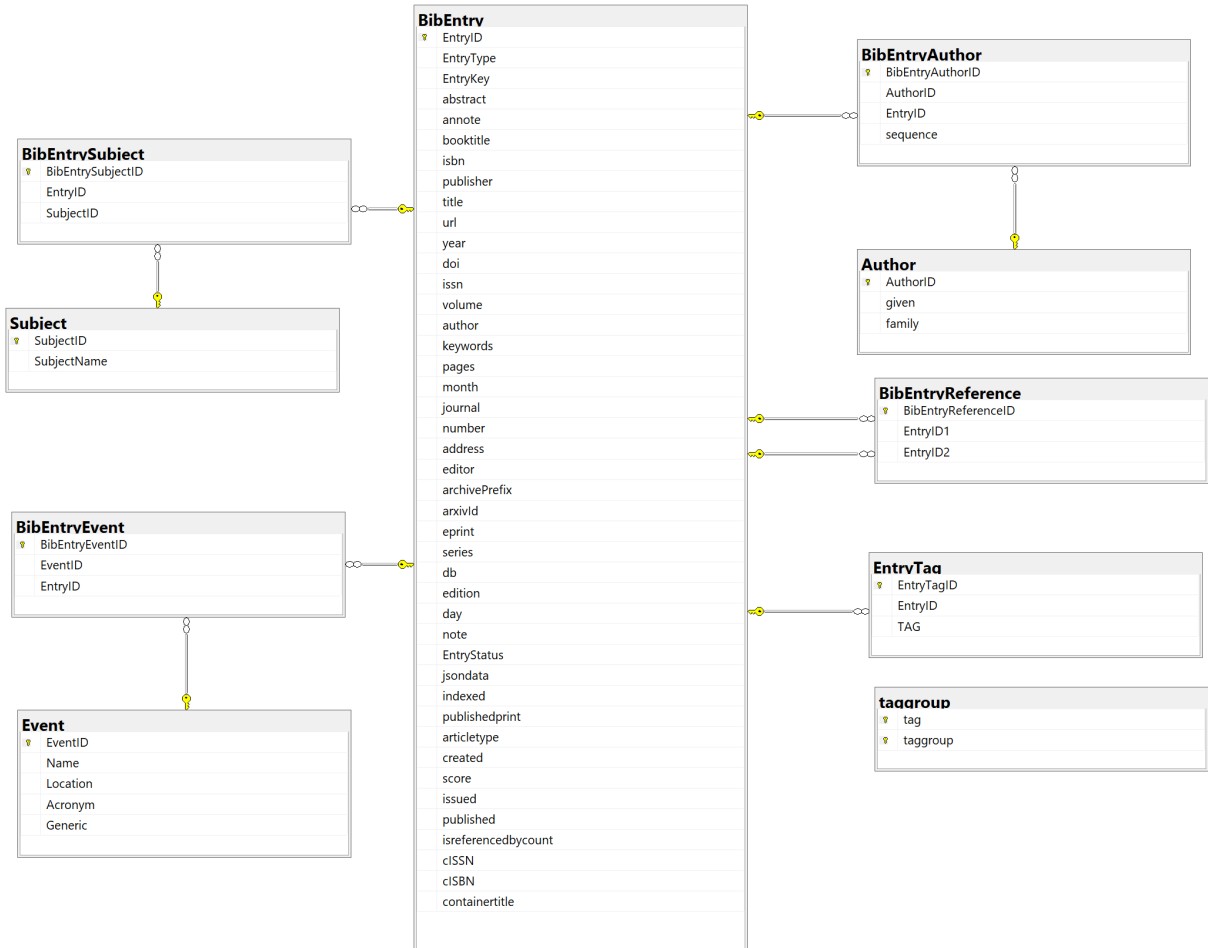

**Figure 11.** SQL tables structure.

### 5.4. Data Enrichment

The information that comes in the BibTeX format provided by scientific databases does not have all the information for processing. For this, it is necessary to carry out a data enrichment process. This process consists of looking for other sources of information for the loaded articles, to fill in empty information or replace it with better information. The objective of this process is to have at least information about the title, authors, subjects, references, and abstract all stored within the SQL database.

The following techniques are performed:

1. DOI to BibTeX Converter (https://www.bibtex.com/c/doi-to-bibtex-converter (accessed on 22 December 2022))

   The objective is to obtain a different source of the article's information and add it to the existing one. This is performed by using a query in the SQL database for obtaining the DOI column of all loaded data to a plain text file. Then those are loaded to the tool website and finally obtain a consolidated BibTeX file. The enrichment process is done by using a program that reads the generated BibTeX file, searching if the information obtained can be used to fill the blank fields.

2. CrossRef REST API (https://api.crossref.org/swagger-ui/index.html (accessed on 22 December 2022))

   Crossref is the official registration agency for managing the digital object identifier (DOI). They have published a Crossref Unified Resource API, in which it is possible to query articles' metadata using REST protocol and JSON serialization (there are other

formats available). They receive the DOI number as a parameter and return detailed information.

For this, a program was written which iterates one by one all the articles registered in previous steps, so to obtain detailed information on each. The algorithm saves the information for the empty fields and for fields that seem to be more confident compared with previous ones.

3.  Mendeley update reference tool

    Mendeley is a references manager developed by Elsevier. This tool helps to manage citations and bibliographies, mainly for research works. Version Mendeley Desktop 1.13 (https://blog.mendeley.com/tag/new-release/ (accessed on 22 December 2022)) added the option called "Update Details", which queries the Mendeley Catalog, searching the last information about an article. It is common to fill only the field DOI in the form and using this tool, Mendeley queries the other fields from Mendeley central library, bringing complementary information.

    For this case, it is generated a BibTeX file from the SQL database which contains only the DOI field filled. Then the file is loaded into Mendeley, and then the "Update Details" option is used to query the data for all the articles considered. Finally, it is needed to export information from Mendeley in BibTeX format. This file is read from a program that compares and fills the information to the empty fields.

At the end of this process, it is expected to have the majority of fields filled into the main SQL table, so the data processing phase can be carried out. There could be several other sources for performing the enrichment process, but in this work, those ones were mainly used.

### 5.5. Acceptance Criteria

Some processes are carried out for applying acceptance criteria, by identifying the articles that will not be considered for the analysis. The following queries are done using SQL language, defining different statuses (field EntryStatus) and marking them to not be considered further.

- Articles without abstract;
- Articles without a DOI number;
- Articles which are duplicated between databases compared by DOI;
- Articles which are duplicated between databases compared by title;
- Articles which does not have complete information;
- Articles which are published in languages different from English.

### 5.6. Article Tagging

A tag in this context refers to a word or set of words used to determine the presence of a specialty, orientation, or direction of research within an article.

This process consists of matching articles with tags available within the collected content. The following steps are followed:

1.  Base List

    This process consists of the author, based on a quick exploratory review, making a preliminary analysis of the specialties, sub-areas, and variants that are published in relation to the root topic, and identifying the possible keywords that would specifically segment them. A base list is created.

2.  Refinement 1 A word cloud is made on the titles or abstracts of the articles, in which the connectors, articles, adverbs, or interjections that do not have the relevance of individual meaning are removed. The graph or the frequency table is analyzed to take the most recurring words. The found ones are added to the list.

3.  Refinement 2 Collateral terms referring to techniques, methods, platforms, or technologies that may be slightly related to the root topic are identified. Those identified are added to the list.

4.  Refinement 3 Articles from other research areas that use a substitute, complementary, or opposite ideas in relation to the root topic are identified. From them, the main terms that determine their variants are identified. Those terms are added to the list.

5.  Refinement 4 The titles and quick review of the abstracts of the articles chosen for the study are reviewed, in search of terms that identify an application, case, or approach. The found terms are added to the list.

6.  Cleaning
    Repeated terms are removed. Terms that are based on very short acronyms, where a textual search could match with a substring within a word, are removed. The terms used in the main search strings of the study are incorporated.

7.  Tag relationship A procedure is executed in the database through SQL queries, in which the list of identified terms is stored in a temporary table and compared with the texts collected from the articles that are part of the study. The information is searched both in the title and in the abstract, by means of a textual search of each term within the text. An article can have one or several related tags, which are recorded in the EntryTag relationship table.

### 5.7. Concept Identification

The tags that have the highest frequency are identified and terms that group them together behind a concept are searched for, generating classification terms.

Through SQL queries, the tag table is also filled with the classification terms (EntryTag) and the relationship table between the tag and the classification term (taggroup).

It will be possible to carry out other types of processing, data enrichment, or generation of temporary tables not listed here, to facilitate the presentation of reports and the interpretation of the results.

### 5.8. Reports Generation

The following types of information summary tables can be considered:

- Distribution of articles by years;
- Distribution of articles by type of article;
- Distribution of articles by journal or conference of origin;
- Distribution of articles by scientific databases of origin;
- Identification of the articles most referred to by other articles;
- Identification of the most used journals on the subject;
- Identification of the most used conferences on the subject;
- Distribution of the most matching journal topics;
- List of classification terms;
- Comprehensive mapping chart.

Other types of tables can be added, as long as the researcher finds better ways to cross-reference the available information; for instance, to join various concepts in tables looking for a new interpretation, using the relational and processing capabilities of the SQL language.

### 5.9. Graphs Generation

One way to present the resulting information is with the use of visual representation graphics. The following types of charts can be considered:

- Pie chart, bar chart, or timeline for distribution of articles by years;
- Pie chart or bar chart for distribution by type of article;
- Pie chart or bar chart for distribution by type of publication (journal or conference);
- Pie chart or bar chart for distribution by scientific databases of origin;
- Timeline of posts on the topic with trend line;
- Tree-Map for visualization in areas of the publication journals;

- Pie chart or bar chart to represent the frequency of use of classification terms or classification groups;
- Correlation graph between variables and determination of correlation indicators.

To generate the indicated graphs, various statistical or graphing tools may be used, which allow the presentation of the information as the researcher considers an adequate form of presentation. The researcher can add other graphics that he deems appropriate for the presentation of the results.

### 5.10. Result Interpretation

Once the information generated is available, the researcher will be able to interpret the results. The SQL database can be used to delve into obtaining specific data that supports it to generate answers to the research questions and objectives of the study. Finally, the writing of the article and publication will be carried out, as appropriate.

## 6. Discussion

### 6.1. The Conclusion Process

To reach the conclusions of this study, a systematic and sequential review of the resulting data was carried out in the data analysis stage, taking note of the most important findings found.

The quality and coherence of the results found have been verified. The research questions have been reviewed one by one, to ensure that the necessary guidelines are available to propose supported answers.

It is assumed that the SQL-based processing method proposed in this paper is a valid approach for determining the findings and the results presented. It is known that using a tags-based classification method may not be precise to determine sub-areas and fine-grain approaches, but is considered applicable for a global classification of a big topic.

### 6.2. Report-Writing Process

The writing process of this article focused on consolidating all the information used during the study, in a systematic way, and preparing it for the journal format.

### 6.3. Discussion about Research Questions

- RQ1: What are the descriptive statistics of the publications among selected articles?
  As seen in the data analysis section, it has been possible to make a broad statistical description of the studies selected for this study. From this it is possible to highlight the following:

  – A total of 683 articles were selected, considered as the primary studies or the ones that met the search chains and applicable criteria.
  – The terms most found in the analysis, which refer to a particular research area, correspond to the following: performance, parallel, architecture, optimization, software, hardware, development, memory, GPU, and test.
  – As mentioned in Figure 5, most of the articles come from the Scopus database, after removing duplicates.
  – In Table 6 it is possible to see the articles that refer to transpilers, which have the largest number of references.
  – Figure 6 and the Equation (1) show that the trend of publications regarding transpilers has remained stable. Through a linear regression, it has been possible to see that the trend is slightly downward, which indicates that it is expected that there will be publications on the subject in a stable manner.
  – According to Figure 7, it has been possible to see that practically half of the selected articles have been published in scientific journals and the other half in scientific conferences.
  – In Table 8 it is possible to see the journals most used for publications related to transpilers.

- In Table 9 it is possible to see the conferences most used for publications related to transpilers.
- In Tables 10 and 11 it is possible to see the most recurrent authors in the publication of articles in relation to transpilers.
- In Table 13 it is possible to see the relationship of greater co-occurrence of terms in the same articles, allowing the determination of the most common research areas.
- In Table 15, it is possible to see the programming languages most used in relation to transpilers.
- Table 16 shows the most commonly used transpiler synonyms in scientific articles.

- RQ2: In which scenarios transpilers are most used?
  Referring to Table 14, the major research areas in relation to the use of transpilers can be evidenced. From this it has been possible to see that the scenarios where transpilers are mostly used are the following:

  - To improve performance in algorithms related to high-performance computing (HPC).
  - For preparation for parallel execution, mainly in relation to the transformation of sequential methods to algorithms prepared to take advantage of the GPU, CUDA cores, OpenMP, and others.
  - For the creation of automatic transformations between programming languages commonly used in the industry.
  - As preparation of methods to support different processor architectures.
  - As part of the internal procedures of a compiler and its execution platforms.
  - As an element to facilitate the processes of diagnosis and execution of software testing exercises.
  - To improve graphic processing procedures.
  - To generate compatibility with embedded systems so that the common code can be transferred to the specific languages of the specific circuits.
  - As part of the source code generation architecture.
  - As a constituent element in artificial intelligence methods.
  - As a base for the transformation of code for preparation of execution of the front-end of applications.
  - As part of the compatibility processes with the different mobile platforms.
  - As a method to transform sentences between parameterization languages, configuration or platform of network devices.
  - As a means to transfer, transform, or reuse mathematical models.
  - As a compatibility tool for the different platforms for the creation and execution of digital games.

- RQ3: In which kind of industry applications transpilers are commonly used?
  As it is possible to see in Section 4 of this document, the applications in the industry are very varied and of diverse uses. Among the most outstanding in recent times, the following are mainly included:

  - For the direct transformation from one language to another, as a code reuse tool or migration strategy for legacy platforms.
  - As a base language for the operation of a front-end framework, for compatibility with end-user and mobile platforms, for transformations from web language to native code of the platforms.
  - To generate language extensions, mainly as a superset of Javascript.
  - For source code refactoring processes.

- RQ4: Which programming languages and technologies are mainly related to transpilers?
  In Table 15 it is possible to see the programming languages that are mostly related to transpilers. In this context, it is possible to see that the relationship with the C family

of languages stands out. Below are Java language and Javascript as the most-referred languages.

- RQ5: Which are the usages of transpilers in the scope of the back-end of a transactional application?

  The use of transpilers has been revised for the specific area of transactional applications. The front-end has had the greatest emphasis in this type of study, through the proposal of frameworks that transform web code into native mobile code or to have languages with greater capacities and expressiveness, and in the end, transform them into basic languages of execution. We could not find a framework that involves a transpiler in the back-end component.

- Regarding the proposed SQL-Based processing method, This method is a new approach to the qualitative processing of literature reviews. There are some tools that use similar conceptions of processing, for example, the Autocoding tool of Atlas.ti (https://atlasti.com/research-hub/auto-coding-and-smart-coding-in-atlas-ti-web (accessed on 22 December 2022)). The difference is that the proposed one uses commonly accessible tools for researchers and its capability of making ad hoc queries to deep inside the retrieved information.

### 6.4. Threats to Validity

#### 6.4.1. Internal Validity

Internal validity in this case is defined as a method to analyze the internal part of the present study. It seeks to demonstrate how robust the proposed methodology is, and the elements that may have introduced some degree of bias.

In this sense and as mitigation, a very detailed description of the methodology and methods applied, and of the steps followed during the information processing, has been presented.

The elements that are identified as possible threats to internal validity are the following:

- The SQL-based processing method proposed is a new way to process literature reviews using a qualitative approach. The conception of this method allows for finding research areas based on tags found in the text. This raises the problem that a term can be used in different contexts and ways. The validity of this method is based on the fact that, although it is not precise, the greater the appearance of certain terms in the articles, the better it will allow for finding tendencies and similarities and be certain of those that are mostly repeated, which provides a possible identification of a common theme and dialectic between researchers in the area. The idea is that by statistical tendency and the law of large numbers, it is possible to infer certain research areas that have multiple terms in common. The level of detail is sacrificed in exchange for achieving a broad-spectrum mapping, which, when having to review a large number of articles, is useful to later move on to a detailed review if needed.
- The selection of the scientific databases could not involve any important one that had information of focus in the computer sciences.
- Although the corresponding mitigation was carried out through a sequential and strict exercise, there could be possible differences and biases incorporated at the time of the creation of the search strings and their execution in the scientific databases.
- Although the corresponding mitigation was made through data processing techniques, there could be differences and biases regarding the selection of classification terms. Others might have been missing that the method used might not have been considered, but that may have been important.

#### 6.4.2. External Validity

External validity in this case seeks to determine how generalizable the results presented are.

Due to the fact that the present study presents a broad mapping of the literature, in which all the applications related to transpilers are searched without a specific filter, it

can be considered that the process was carried out toward the complete universe of this research branch, according to the established design criteria. That is why a certain high level of generalizability can be mentioned in the results.

## 7. Conclusions

### 7.1. Study Conclusions

Transpilers refer to a technology that allows translating source code made with a source programming language into source code made with a target programming language. This technique has been used in many different applications in academia and industry. It has been considered important to analyze the extent, scope, and nature of this type of study, in order to determine the specific areas, approaches, and topics where it is being used most, in addition to identifying possible patterns or common designs where it is used in the field of software architecture design.

Various research questions were established, related to answering questions of identification of the research areas and the descriptive statistics of the related articles.

A research method called PSALSAR was used which follows the following steps: protocol, for defining study scope; search, for defining search strategy; appraisal, for selecting the studies; synthesis, for extracting and categorizing data; analysis, for results formulation; and report, for writing the report.

Four scientific databases were selected. The search strings were designed and adapted for execution in the selected scientific databases. The searches were carried out and the inclusion and exclusion criteria were applied. With the primary studies determined, the data collection process was carried out using the SQL-based processing method described in this document, which uses a relational database with the aim of facilitating and automating the tabulation process of the collected data.

Consolidation information of the results was generated and the corresponding graphs were generated. Finally, a discussion process was carried out in which the research questions and the established approaches to answer them were analyzed one by one.

In a summary, a systematic mapping review has been successfully carried out, which allows a comprehensive mapping on the topic of transpilers using a search string oriented to process the entire universe of research on the selected topic.

The approach of an automatic data processing method based on personalized programs and a relational database, which facilitates this type of process, has been carried out.

A review of the applicability of transpilers in the industry has been carried out. Several technologies, focused mainly on front-end development, have been identified. Transpilers are widely used inside frameworks for supporting multi-platform targeting. There appears to be no reference to a transpiler-based proposal focused on the back-end of transactional applications. The most used target programming languages in relation to a transpiler are C++ / C, Java, and Javascript.

The main research areas have been identified within the framework of transpilers. Table 14 presents a mapping table which allows visualizing the topics that are mostly presented in the studies and their references. We can see that performance and parallel computing, referring to different compilation or refactoring procedures, are the topics most covered using transpilers in the research field. It is possible to identify also different kinds of applications for embedded systems, mobile, and support for different microcontrollers and peripherals that use transpilers to have a way to write only once but support the execution in different platforms using native code. Transpilers are tools that still represent a broad and constant topic in research or as a tool for enabling the development of new kinds of technologies or research projects, which will continue presenting different applications and usage topics.

The SQL-based processing method represents a new way of processing a large number of articles for conducting a high-level mapping review. It is based on identifying common tags inside articles, by processing text using SQL commands, and then classifying them into general concepts. Despite this method being still in the process of validation, during

this work it was considered useful because it allowed us to obtain valuable information. It was possible to identify the main referenced articles by topic, the most used journals and conferences, the trending articles published through the years, and the main concepts that researchers are actively working last years. By using this method and in combination with complementary methods, it was possible to answer all the research questions proposed, which are detailed in the discussion section.

*7.2. Future Work*

The execution of future update studies, to determine the articles and new research trends in future years.

The presentation of SQL-based processing methods presented in the present study as an independent, detailed article with elements of greater application detail, where the present study could be the first example of its applicability.

There are research areas where not such intensive use has been made, and which offer the possibility of opening new studies, as well as for example for the back-end in transactional software, migration strategies for legacy systems, AI, math, multiplatform games, apps, automatic source code generation, and networking.

**Author Contributions:** Conceptualization and Execution, A.B.F.; methodology, A.B.F., M.P. and J.M.H.; validation, M.P. and J.M.H.; supervision, M.P.; project administration, A.B.F.; funding acquisition, A.B.F. and M.P. All authors have read and agreed to the published version of the manuscript.

**Funding:** This research was partially funded by Smartwork S.A. company.

**Institutional Review Board Statement:** Not applicable.

**Informed Consent Statement:** Not applicable.

**Data Availability Statement:** Not applicable.

**Conflicts of Interest:** The authors declare no conflicts of interest.

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
