# Peer review of "Transpilers: A Systematic Mapping Review of Their Usage in Research and Industry"

_applsci, doi:10.3390/app13063667_

Round 1

Reviewer 1 Report

The paper presents a comprehensive systematic review of the implementation and use of transpilers in research studies in the last ten years. The manuscript is easy to read and provides an adequate description of the review methodology and analysis. The review methodology mainly depends on the use of SQL-based processing of metadata associated with the reviewed publications. Besides, a utility is developed to extract and transfer this information into the SQL database. The systematic review is conducted to address several research questions, which in my opinion, need further In-depth investigation and analysis. The first research question might be attainable considering using a qualitative tabulation tool, which generates non-semantic categorization of metadata. However, the remaining four questions are challenging to address, considering only the SQL-based analysis. Please provide further details about the process of answering these questions by linking papers with categories supporting your conclusions. Figures 3, 4, and Table 5 present the list of words, terms, and tags extracted from reviewed publications. These terms may not present evidence to classify the set of retrieved publications. These tags and terms are general and could be linked to different categorizations. Even after cleaning and linking to abstracts, titles, and journals, the generated categorization is still vague. Table 14 shows the relation between the main concepts and the list of related tags. It is not clear in the manuscript how this table was created. The authors mention that they conduct a complimentary review to identify the most recurring global concepts in the use of transpilers. However, they should provide more details about the methodology of this review and explain how and why some of the selected articles are chosen to provide evidence connecting tags with concepts.

Figure 5 illustrates the distribution of articles per scientific database. Yet, the x-axis of the Figure starts at 75%, which is visually misleading since readers may not see the difference between the scale of the three scientific databases.

Figure 6 is an Equation used to compute the trendline for the number of articles per year. This equation need not be presented as a Figure.  

There are a few grammar mistakes in the manuscript, like the one in the first line of the introduction; please proofread the paper. 

Reviewer 2 Report

How Block1. Motivation should be added in the abstract section.

2. Literature should be enhanced for better understanding.

3. Some technical RQ should be added.

4. RQ should answer in a better way for more clarity.

5. Conclusion section should be re-drafted with the outcome of the work.

6. Some background in technical aspects should also be covered in the research article.

7. Few articles should be added to enhance the literature abs application. Of work :

- Empirical Study of Software Defect Prediction: A Systematic Mapping. Symmetry. 2019; 11(2):212. https://doi.org/10.3390/sym11020212

- An extensive evaluation of search-based software testing: a review. Soft Comput 23, 1933–1946 (2019). https://doi.org/10.1007/s00500-017-2906-y

- "Proliferation of Opportunistic Routing: A Systematic Review," in IEEE Access, vol. 10, pp. 5855-5883, 2022, doi: 10.1109/ACCESS.2021.3136927.chain Technology Can Transfigure the Indian Agriculture Sector

-Sugandh, U., Khari, M., & Nigam, S. (2022). How Blockchain Technology Can Transfigure the Indian Agriculture Sector: A Review. Handbook of Green Computing and Blockchain Technologies, 69-88.

Reviewer 3 Report

Acceptable in the present mode

Reviewer 4 Report

Dear Authors

With the compliments of custom, I congratulate you on your extensive work reviewing the literature on the topic of transpilation process. The work was extensive and very compartmentalized. The theme is exciting and presents contributions to the study area. However, the authors need to conciliate the scientific rigor with the fluidity of the text. In this way, I will make some suggestions for improvement:

1) I believe it is unnecessary to title each abstract partition in the abstract. ("Context:"; "Objectives:"...etc.) I suggest the suppression of these terms;

2) Still in relation to the Abstract, I suggest the insertion of the limitations of the research and future research indications;

3) The introduction is pervasive. I suggest not subdividing it into subsections. I suggest that subsection 1.3 be inserted in the Materials and Methods; subsections 2.1.1. and 2.1.2 be inserted in the introduction before the paragraph of line 132.

4) Figures should come before the comments about them.

5) The formula illustrated in figure 6 should be written and not illustrated.

6) Figure 6 is placed before figure 5.

7) Figure 5 would be better represented if it were transformed into a table.

8) The authors should review the position of the comments about tables and figures because the insertions are not occurring logically. For example, in line 340, the authors cite figure 7, and in line 343, they cite figure 6. However, figure 7 cited should be figure 5.

9) Tables 6, 7, 12, and 14, I suggest inserting them in a landscape layout sheet.

10) Figure 7 and 8 with low quality. I suggest inserting them in the landscape layout.

11) I suggest reducing the footnotes. Only place them if the information is relevant.

Good review.

Revisor

Round 2

Reviewer 4 Report

Dear Authors

Initially, I congratulate the authors for the extensive revision work and attention to the suggestions proposed by the reviewers. In the current version, I believe the article meets the minimum conditions for acceptance for publication.

Best Regards

Reviewer